# Memory-efficient Reinforcement Learning with Value-based Knowledge Consolidation

**Qingfeng Lan** [*]                                                                qlan3@ualberta.ca
*Department of Computing Science*
*University of Alberta*

**Yangchen Pan** [†]                                                        yangchen.pan@eng.ox.ac.uk
*University of Oxford*

**Jun Luo**                                                                       jun.luo1@huawei.com
*Huawei Noah's Ark Lab*

**A. Rupam Mahmood**                                                      armahmood@ualberta.ca
*Department of Computing Science*
*University of Alberta*
*CIFAR AI Chair, Amii*

**Reviewed on OpenReview:** *https://openreview.net/forum?id=zSDCvlaVBn*

## Abstract

Artificial neural networks are promising for general function approximation but challenging to train on non-independent or non-identically distributed data due to catastrophic forgetting. The experience replay buffer, a standard component in deep reinforcement learning, is often used to reduce forgetting and improve sample efficiency by storing experiences in a large buffer and using them for training later. However, a large replay buffer results in a heavy memory burden, especially for onboard and edge devices with limited memory capacities. We propose memory-efficient reinforcement learning algorithms based on the deep Q-network algorithm to alleviate this problem. Our algorithms reduce forgetting and maintain high sample efficiency by consolidating knowledge from the target Q-network to the current Q-network. Compared to baseline methods, our algorithms achieve comparable or better performance in both feature-based and image-based tasks while easing the burden of large experience replay buffers. [1]

## 1 Introduction

When trained on non-independent and non-identically distributed (non-IID) data and optimized with stochastic gradient descent (SGD) algorithms, neural networks tend to forget prior knowledge abruptly, a phenomenon known as *catastrophic forgetting* (French, 1999; McCloskey & Cohen, 1989). It is widely observed in continual supervised learning (Hsu et al., 2018; Farquhar & Gal, 2018; Van de Ven & Tolias, 2019; Delange et al., 2021) and reinforcement learning (RL) (Schwarz et al., 2018; Khetarpal et al., 2020; Atkinson et al., 2021). In the RL case, an agent receives a non-IID stream of experience due to changes in policy, state distribution, the environment dynamics, or simply due to the inherent structure of the environment (Alt et al., 2019). This leads to catastrophic forgetting that previous learning is overridden by later training,

---

[*]Work partially done as an intern at Noah's Ark Lab, Huawei Canada.
[†]Work done while at Noah's Ark Lab, Huawei Canada.

[1]Code release: `https://github.com/qlan3/MeDQN`

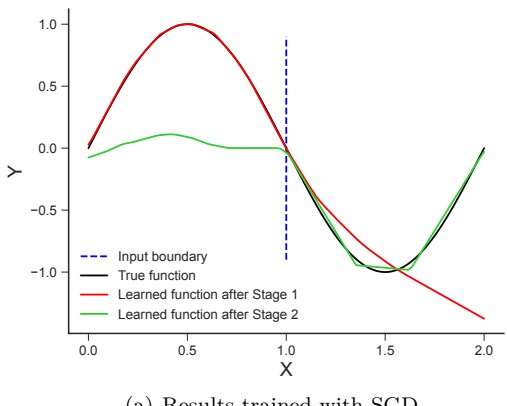
(a) Results trained with SGD.

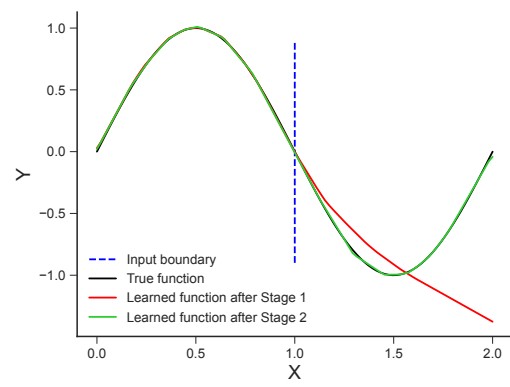
(b) Results trained with SGD and knowledge consolidation (our method).

Figure 1: A visualization of learned functions after training as well as the true function. In Stage 1, we generate training samples $(x, y)$, where $x \in [0, 1]$ and $y = \sin(\pi x)$. In Stage 2, $x \in [1, 2]$ and $y = \sin(\pi x)$. The blue dotted line shows the boundary of the input space for Stage 1 and 2.

resulting in deteriorating performance during single-task training (Ghiassian et al., 2020; Pan et al., 2022a). To mitigate this problem, Lin (1992) equipped the learning agent with experience replay, storing recent transitions collected by the agent in a buffer for future use. By storing and reusing data, the experience replay buffer alleviates the problem of non-IID data and dramatically boosts sample efficiency and learning stability. In deep RL, the experience replay buffer is a standard component, widely applied in value-based algorithms (Mnih et al., 2013; 2015; van Hasselt et al., 2016), policy gradient methods (Schulman et al., 2015; 2017; Haarnoja et al., 2018; Lillicrap et al., 2016), and model-based methods (Heess et al., 2015; Ha & Schmidhuber, 2018; Schrittwieser et al., 2020).

However, using an experience replay buffer is not an ideal solution to catastrophic forgetting. The memory capacity of the agent is limited by the hardware, while the environment itself may generate an indefinite amount of observations. To store enough information about the environment and get good learning performance, current methods usually require a large replay buffer (e.g., a buffer of a million images). The requirement of a large buffer prevents the application of RL algorithms to the real world since it creates a heavy memory burden, especially for onboard and edge devices (Hayes et al., 2019; Hayes & Kanan, 2022). For example, Wang et al. (2023) showed that the performance of SAC (Haarnoja et al., 2018) decreases significantly with limited memory due to hardware constraints of real robots. Smith et al. (2022) demonstrated that a quadrupedal robot can learn to walk from scratch in 20 minutes in the real-world. However, to be able to train outdoors with enough memory and computation, the authors had to carry a heavy laptop tethered to a legged robot during training, which makes the whole learning process less human-friendly. The world is calling for more memory-efficient RL algorithms.

In this work, we propose memory-efficient RL algorithms based on the deep Q-network (DQN) algorithm. Specifically, we assign a new role to the target neural network, which was introduced originally to stabilize training (Mnih et al., 2015). In our algorithms, the target neural network plays the role of a knowledge keeper and helps consolidate knowledge in the action-value network through a consolidation loss. We also introduce a tuning parameter to balance learning new knowledge and remembering past knowledge. With the experiments in both feature-based and image-based environments, we demonstrate that our algorithms, while using an experience replay buffer at least 10 times smaller compared to the experience replay buffer for DQN, still achieve comparable or even better performance.

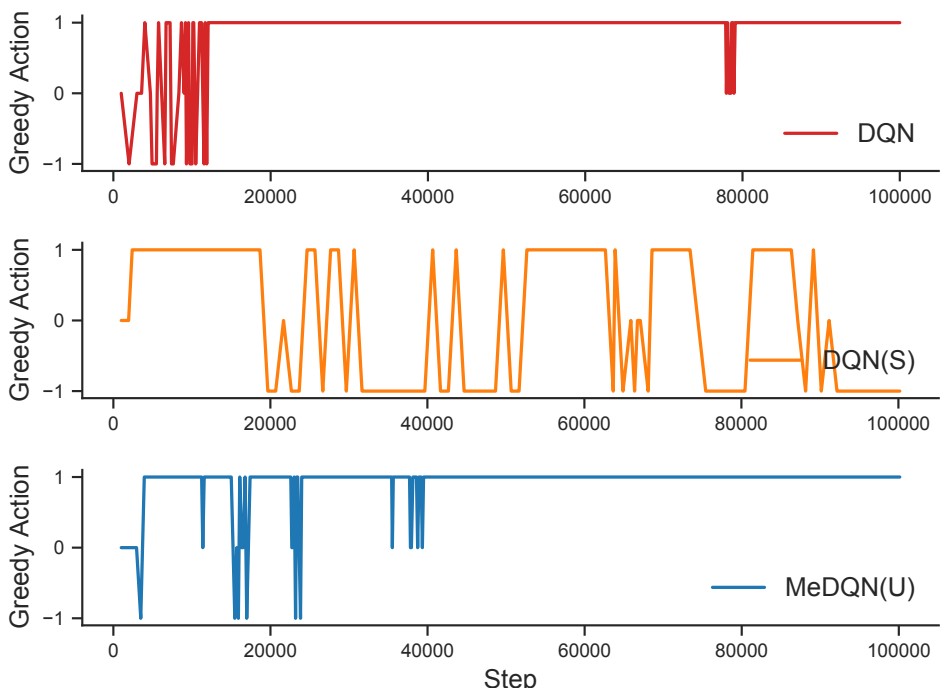

Figure 2: The greedy actions of a randomly sampled state for different methods during training in Mountain Car. There are 3 actions in total, and the optimal action is 1. Without a large replay buffer, DQN(S) keeps forgetting and relearning the optimal action, while DQN and MeDQN(U) suffer much less from forgetting.

## 2 Understanding forgetting from an objective-mismatch perspective

In this section, we first use a simple example in supervised learning to shed some light on catastrophic forgetting from an objective-mismatch perspective. Let $D$ be the whole training dataset. We denote the true objective function on $D$ as $L_D(\theta_t)$, where $\theta_t$ is a set of parameters at time-step $t$. Denote $B_t$ as a subset of $D$ used for training at time-step $t$. We expect to approximate $L_D(\theta_t)$ with $L_{B_t}(\theta_t)$. For example, $B_t$ could be a mini-batch sampled from $D$. It could also be a sequence of temporally correlated samples when samples in $D$ come in a stream. When $B_t$ is IID (e.g., sampling $B_t$ uniformly at random from $D$), there is no objective mismatch since $L_D(\theta_t) = \mathbb{E}_{B_t \sim D}[L_{B_t}(\theta_t)]$ for a specific $t$. However, when $B_t$ is non-IID, it is likely that $L_D(\theta_t) \neq \mathbb{E}_{B_t \sim D}[L_{B_t}(\theta_t)]$, resulting in the objective mismatch problem. The mismatch between optimizing the true objective and optimizing the objective induced by $B_t$ often leads to catastrophic forgetting. Without incorporating additional techniques, catastrophic forgetting is highly likely when SGD algorithms are used to train neural networks given non-IID data — the optimization objective is simply wrong.

To demonstrate how objective mismatch can lead to catastrophic forgetting, we provide a regression experiment. Specifically, a neural network was trained to approximate a sine function $y = \sin(\pi x)$, where $x \in [0, 2]$. To get non-IID input data, we consider two-stage training. In Stage 1, we generated training samples $(x, y)$, where $x \in [0, 1]$ and $y = \sin(\pi x)$. For Stage 2, $x \in [1, 2]$ and $y = \sin(\pi x)$. The network was first trained with samples in Stage 1 in the traditional supervised learning style. After that, we continued training the network in Stage 2. Finally, we plotted the learned function after the end of training for each stage. More details are included in the appendix.

As shown in Figure 1(a), after Stage 1, the learned function fits the true function $y = \sin(\pi x)$ on $x \in [0, 1]$ almost perfectly. At the end of Stage 2, the learned function also approximates the true function on $x \in [1, 2]$ well. However, the network catastrophically forgets function values it learned on $x \in [0, 1]$, due to input distribution shift. As stated above, the objective function we minimize is defined by training samples only from one stage (i.e. $x \in [0, 1]$ or $x \in [1, 2]$) that are not enough to reconstruct the true objective properly

which is defined on the whole training set (i.e. $x \in [0, 2]$). Much of the previously learned knowledge is lost while optimizing a wrong objective.

A similar phenomenon also exists in single RL tasks. Specifically, we show that, without a large replay buffer, DQN can easily forget the optimal action after it has learned it. We tested DQN in Mountain Car (Sutton & Barto, 2018), with and without using a large replay buffer. Denote *DQN(S)* as DQN using a tiny (32) experience replay buffer. We first randomly sampled a state $S = [-0.70167243, 0.04185214]$ and then recorded its greedy action (i.e., $\arg\max_a Q(S, a)$) through the whole training process, as shown in Figure 2. Note that the optimal action for this state is 1. We did many runs to verify the result and only showed the single-run result here for better visualization. We observed that, when using a large buffer (i.e. $10K$), DQN suffered not much from forgetting. However, when using a small buffer (i.e. 32), DQN(S) kept forgetting and relearning the optimal action, demonstrating the forgetting issue in single RL tasks. MeDQN(U) is our algorithm which will be introduced later in Section 4. As shown in Figure 2, MeDQN(U) consistently chose the optimal action 1 as its greedy action and suffered much less from forgetting, even though it also used a tiny replay buffer with size 32. More training details can be found in Section 5.3 and the appendix.

## 3 Related works on reducing catastrophic forgetting

The key to reducing catastrophic forgetting is to preserve past acquired knowledge while acquiring new knowledge. In this section, we will first discuss existing methods for continual supervised learning and then methods for continual RL. Since we can not list all related methods here, we encourage readers to check recent surveys (Khetarpal et al., 2020; Delange et al., 2021).

### 3.1 Supervised learning

In the absence of memory constraints, rehearsal methods, also known as *replay*, are usually considered one of the most effective methods in continual supervised learning (Kemker et al., 2018; Farquhar & Gal, 2018; Van de Ven & Tolias, 2019; Delange et al., 2021). Concretely, these methods retain knowledge explicitly by storing previous training samples (Rebuffi et al., 2017; Riemer et al., 2018; Hayes et al., 2019; Aljundi et al., 2019a; Chaudhry et al., 2019; Jin et al., 2020). In generative replay methods, samples are stored in generative models rather than in a buffer (Shin et al., 2017; Kamra et al., 2017; Van de Ven & Tolias, 2018; Ramapuram et al., 2020; Choi et al., 2021). These methods exploit a dual memory system consisting of a student and a teacher network. The current training samples from a data buffer are first combined with pseudo samples generated from the teacher network and then used to train the student network with knowledge distillation (Hinton et al., 2014). Some previous methods carefully select and assign a subset of weights in a large network to each task (Mallya & Lazebnik, 2018; Sokar et al., 2021; Fernando et al., 2017; Serra et al., 2018; Masana et al., 2021; Li et al., 2019; Yoon et al., 2018) or assign a task-specific network to each task (Rusu et al., 2016; Aljundi et al., 2017). Changing the update rule is another approach to reducing forgetting. Methods such as GEM (Lopez-Paz & Ranzato, 2017), OWM (Zeng et al., 2019), and ODG (Farajtabar et al., 2020) protect obtained knowledge by projecting the current gradient vector onto some constructed space related to previous tasks. Finally, parameter regularization methods (Kirkpatrick et al., 2017; Schwarz et al., 2018; Zenke et al., 2017; Aljundi et al., 2019b) reduce forgetting by encouraging parameters to stay close to their original values with a regularization term so that more important parameters are updated more slowly.

### 3.2 Reinforcement learning

RL tasks are natural playgrounds for continual learning research since the input is a stream of temporally structured transitions (Khetarpal et al., 2020). In continual RL, most works focus on incremental task learning where tasks arrive sequentially with clear task boundaries. For this setting, many methods from continual supervised learning can be directly applied, such as EWC (Kirkpatrick et al., 2017) and MER (Riemer et al., 2018). Many works also exploit similar ideas from continual supervised learning to reduce forgetting in RL. For example, Ammar et al. (2014) and Mendez et al. (2020) assign task-specific parameters to each task while sharing a reusable knowledge base among all tasks. Mendez et al. (2022) and Isele & Cosgun (2018)

use an experience replay buffer to store transitions of previous tasks for knowledge retention. The above methods cannot be applied to our case directly since they require either clear task boundaries or large buffers. The student-teacher dual memory system is also exploited in multi-task RL, inducing behavioral cloning by function regularization between the current network (the student network) and its old version (the teacher network) (Rolnick et al., 2019; Kaplanis et al., 2019; Atkinson et al., 2021). Note that in continual supervised learning, there are also methods that regularize a function directly (Shin et al., 2017; Kamra et al., 2017; Titsias et al., 2020). Our work is inspired by this approach in which the target Q neural network plays the role of the teacher network that regularizes the student network (i.e., current Q neural network) directly.

In single RL tasks, the forgetting issue is under-explored and unaddressed, as the issue is masked by using a large replay buffer. In this work, we aim to develop memory-efficient single-task RL algorithms while achieving high sample efficiency and training performance by reducing catastrophic forgetting.

## 4 MeDQN: Memory-efficient DQN

### 4.1 Background

We consider RL tasks that can be formalized as Markov decision processes (MDPs). Formally, let $M = (\mathcal{S}, \mathcal{A}, \mathrm{P}, r, \gamma)$ be an MDP, where $\mathcal{S}$ is the discrete state space [2], $\mathcal{A}$ is the discrete action space, $\mathrm{P} : \mathcal{S} \times \mathcal{A} \times \mathcal{S} \to [0, 1]$ is the state transition probability function, $r : \mathcal{S} \times \mathcal{A} \to \mathbb{R}$ is the reward function, and $\gamma \in [0, 1)$ is the discount factor. For a given MDP, the agent interacts with the MDP according to its policy $\pi$ which maps a state to a distribution over the action space. At each time-step $t$, the agent observes a state $S_t \in \mathcal{S}$ and samples an action $A_t$ from $\pi(\cdot|S_t)$. Then it observes the next state $S_{t+1} \in \mathcal{S}$ according to the transition function $\mathrm{P}$ and receives a scalar reward $R_{t+1} = r(S_t, A_t) \in \mathbb{R}$. Considering an episodic task with horizon $T$, we define the return $G_t$ as the sum of discounted rewards, that is, $G_t = \sum_{k=t}^{T} \gamma^{k-t} R_{k+1}$. The action-value function of policy $\pi$ is defined as $q_\pi(s, a) = \mathbb{E}[G_t|S_t = s, A_t = a], \forall (s, a) \in \mathcal{S} \times \mathcal{A}$. The agent's goal is to find an optimal policy $\pi$ that maximizes the expected return starting from some initial states.

As an off-policy algorithm, Q-learning (Watkins, 1989) is an effective method that aims to learn the action-value function estimate $Q : \mathcal{S} \times \mathcal{A} \to \mathbb{R}$ of an optimal policy. Mnih et al. (2015) propose DQN, which uses a neural network to approximate the action-value function. The complete algorithm is presented as Algorithm 2 in the appendix. Specifically, to mitigate the problem of divergence using function approximators such as neural networks, a target network $\hat{Q}$ is introduced. The target neural network $\hat{Q}$ is a copy of the Q-network and is updated periodically for every $C_{target}$ steps. Between two individual updates, $\hat{Q}$ is fixed. Note that this idea is not limited to value-based methods; the target network is also used in policy gradient methods, such as SAC (Haarnoja et al., 2018) and TD3 (Fujimoto et al., 2018).

### 4.2 Knowledge consolidation

Originally, Hinton et al. (2014) proposed distillation to transfer knowledge between different neural networks effectively. Here we refer to knowledge consolidation as a special case of distillation that transfers information from an old copy of this network (e.g. the target network $\hat{Q}$, parameterized by $\theta^-$) to the network itself (e.g. the current network $Q$, parameterized by $\theta$), consolidating the knowledge that is already contained in the network. Unlike methods like EWC (Kirkpatrick et al., 2017) and SI (Zenke et al., 2017) that regularize parameters, knowledge consolidation regularizes the function directly. Formally, we define the (vanilla) consolidation loss as follows:

$$L_{consolid}^{V} \doteq \mathbb{E}_{(S,A) \sim p(\cdot, \cdot)} \left[ \left( Q(S, A; \theta) - \hat{Q}(S, A; \theta^-) \right)^2 \right],$$

where $p(s, a)$ is a sampling distribution over the state-action pair $s, a$. To retain knowledge, the state-action space should be covered by $p(s, a)$ sufficiently, such as $p(s, a) \doteq d^\pi(s)\pi(a|s)$ or $p(s, a) \doteq d^\pi(s)\mu(a)$, where $\pi$ is

---

[2]We consider discrete state spaces for simplicity. In general, our method also works with continuous state spaces.

the $\epsilon$-greedy policy, $d^\pi$ is the stationary state distribution of $\pi$, and $\mu$ is a uniform distribution over $\mathcal{A}$. Our preliminary experiment shows that $p(s, a) = d^\pi(s)\mu(a)$ is a better choice compared with $p(s, a) = d^\pi(s)\pi(a|s)$; so we use $p(s, a) = d^\pi(s)\mu(a)$ in this work. However, the optimal form of $p(s, a)$ remains an open problem and we leave it for future research. To summarize, the following consolidation loss is used in this work:

$$L_{consolid} \doteq \mathbb{E}_{S \sim d^\pi} \left[ \sum_{A \in \mathcal{A}} \left( Q(S, A; \theta) - \hat{Q}(S, A; \theta^-) \right)^2 \right], \tag{1}$$

Intuitively, minimizing the consolidation loss can preserve previously learned knowledge by penalizing $Q(s, a; \theta)$ for deviating from $\hat{Q}(s, a; \theta^-)$ too much. In general, we may also use other loss functions, such as the Kullback–Leibler (KL) divergence $D_{\mathrm{KL}}(\pi(\cdot|s)||\hat{\pi}(\cdot|s))$, where $\pi$ and $\hat{\pi}$ can be softmax policies induced by action values, that is, $\pi(a|s) \propto \exp(Q(s, a; \theta))$ and $\hat{\pi}(a|s) \propto \exp(Q(s, a; \theta^-))$. For simplicity, we use the mean squared error loss, which also proves to be effective as shown in our experiments.

Given a mini-batch $B$ consisting of transitions $\tau = (s, a, r, s')$, the DQN loss is defined as

$$L_{DQN} = \sum_{\tau \in B} \left( r + \gamma \max_{a'} \hat{Q}(s', a'; \theta^-) - Q(s, a; \theta) \right)^2.$$

We combine the two losses to obtain the final training loss for our algorithms

$$L = L_{DQN} + \lambda L_{consolid},$$

where $\lambda$ is a positive scalar. No extra network is introduced since the target network is used for consolidation. Note that $L_{DQN}$ helps $Q$ network learn **new** knowledge from $B$ that is sampled from the experience replay buffer. In contrast, $L_{consolid}$ is used to preserve **old** knowledge by consolidating information from $\hat{Q}$ to $Q$. By combining them with a weighting parameter $\lambda$, we balance learning and preserving knowledge simultaneously. Moreover, since $L_{consolid}$ acts as a functional regularizer, the parameter $\theta$ may change significantly as long as the function values $Q$ remain close to $\hat{Q}$.

There is still one problem left: how to get $d^\pi(s)$? In general, it is hard to compute the exact form of $d^\pi(s)$. Instead, we use random sampling, as we will show next.

### 4.3  Uniform state sampling

One of the simplest ways to approximate $d^\pi(s)$ is with a uniform distribution on $\mathcal{S}$. Formally, we define this version of consolidation loss as

$$L^U_{consolid} \doteq \mathbb{E}_{S \sim U} \left[ \sum_{A \in \mathcal{A}} \left( Q(S, A; \theta) - \hat{Q}(S, A; \theta^-) \right)^2 \right],$$

where $U$ is a uniform distribution over the state space $\mathcal{S}$. Assuming that the size of the state space $|\mathcal{S}|$ is finite, we have $\Pr(S = s) = 1/|\mathcal{S}|$ for any $s \in \mathcal{S}$. Together with $d^\pi(s) \leq 1$, we then have

$$
\begin{aligned}
L_{consolid} &= \sum_{s \in \mathcal{S}} d^\pi(s) \sum_{a \in \mathcal{A}} \left( Q(s, a; \theta) - \hat{Q}(s, a; \theta^-) \right)^2 \\
&\leq \sum_{s \in \mathcal{S}} \sum_{a \in \mathcal{A}} \left( Q(s, a; \theta) - \hat{Q}(s, a; \theta^-) \right)^2 \\
&= |\mathcal{S}| L^U_{consolid}.
\end{aligned}
\tag{2}
$$

Essentially, minimizing $|\mathcal{S}| L^U_{consolid}$ minimizes an upper bound of $L_{consolid}$. As long as $L^U_{consolid}$ is small enough, we can achieve good knowledge consolidation with a low consolidation loss $L_{consolid}$. In the extreme case, $L^U_{consolid} = 0$ leads to $L_{consolid} = 0$.

In practice, we may not know $\mathcal{S}$ in advance. To solve this problem, we maintain state bounds $s_{LOW}$ and $s_{HIGH}$ as the lower and upper bounds of all observed states, respectively. Note that both $s_{LOW}$ and

---

**Algorithm 1** Memory-efficient DQN with uniform state sampling

---
1: Initialize a small experience replay buffer $D$ (e.g., one mini-batch size)
2: Initialize state lower bound $s_{LOW}$ to $[\infty, \cdots, \infty]$ and upper bound $s_{HIGH}$ to $[-\infty, \cdots, -\infty]$
3: Initialize action-value function $Q$ with random weights $\theta$
4: Initialize target action-value function $\hat{Q}$ with random weights $\theta^- = \theta$
5: Observe initial state $s$
6: **while** agent is interacting with the environment **do**
7:     Update state bounds: $s_{LOW} = \min(s_{LOW}, s)$, $s_{HIGH} = \max(s_{HIGH}, s)$
8:     Take action $a$ chosen by $\epsilon$-greedy based on $Q$, observe $r$, $s'$
9:     Store transition $(s, a, r, s')$ in $D$ and update state $s \leftarrow s'$
10:     **for** every $C_{current}$ steps **do**
11:       Get all transitions ($\approx$ one mini-batch) $(s_B, a_B, r_B, s'_B)$ in $D$
12:       **for** $i = 1$ **to** $E$ **do**
13:         Compute DQN loss $L_{DQN}$
14:         Sample a random mini-batch states $s_{B'}$ uniformly from $[s_{LOW}, s_{HIGH}]$
15:         Compute consolidation loss $L_{consolid}^U$ given $s_{B'}$
16:         Compute the final training loss: $L = L_{DQN} + \lambda L_{consolid}^U$
17:         Perform a gradient descent step on $L$ with respect to $\theta$
18:       **end for**
19:     **end for**
20:     Reset $\hat{Q} = Q$ for every $C_{target}$ steps
21: **end while**

---

$s_{HIGH}$ are two state vectors with the same dimension as a state in $\mathcal{S}$. Assume $\mathcal{S} \subseteq \mathbb{R}^n$. Initially, we set $s_{LOW} = [\infty, \cdots, \infty] \in \mathbb{R}^n$ and $s_{HIGH} = [-\infty, \cdots, -\infty] \in \mathbb{R}^n$. For each newly received $s \in \mathbb{R}^n$, we update state bounds with

$$s_{LOW} = \min(s_{LOW}, s) \text{ and } s_{HIGH} = \max(s_{HIGH}, s).$$

Here, both min and max are element-wise operations. During training, we sample pseudo-states uniformly from the interval $[s_{LOW}, s_{HIGH}]$ to help compute the consolidation loss.

We name our algorithm that uses uniform state sampling as memory-efficient DQN with uniform state sampling, denoted as *MeDQN(U)* and shown in Algorithm 1. Compared with DQN, MeDQN(U) has several changes. First, the experience replay buffer $D$ is extremely small (Line 1). In practice, we set the buffer size to the mini-batch size to apply mini-batch gradient descent. Second, we maintain state bounds and update them at every step (Line 7). Moreover, to extract as much information from a small replay buffer, we use the same data to train the $Q$ function for $E$ times (Line 13–19). In practice, we find that a small $E$ (e.g.,1–4) is enough to perform well. Finally, we apply knowledge consolidation by adding a consolidation loss to the DQN loss as the final training loss (Line 16–17).

## 4.4 Real state sampling

When the state space $\mathcal{S}$ is super large, the agent is unlikely to visit every state in $\mathcal{S}$. Thus, for a policy $\pi$, the visited state set $\mathcal{S}^\pi := \{s \in \mathcal{S} | d^\pi(s) > 0\}$ is expected to be a very small subset of $\mathcal{S}$. In this case, a uniform distribution over $\mathcal{S}$ is far from a good estimation of $d^\pi$. A small number of states (e.g., one mini-batch) generated from uniform state sampling cannot cover $\mathcal{S}^\pi$ well enough, resulting in poor knowledge consolidation for the $Q$ function. In other words, we may still forget previously learned knowledge (i.e. the action values over $\mathcal{S}^\pi \times \mathcal{A}$) catastrophically. This intuition can also be understood from the view of upper bound minimization. In Equation 2, as the upper bound of $L_{consolid}$, $|\mathcal{S}|L_{consolid}^U$ is large when $\mathcal{S}$ is large. Even if $L_{consolid}^U$ is minimized to a small value, the upper bound $|\mathcal{S}|L_{consolid}^U$ may still be too large, leading to poor consolidation.

To overcome the shortcoming of uniform state sampling, we propose real state sampling. Specifically, previously observed states are stored in a state replay buffer $D_s$ and real states are sampled from $D_s$ for

knowledge consolidation. Compared with uniform state sampling, states sampled from a state replay buffer have a larger overlap with $\mathcal{S}^\pi$, acting as a better approximation of $d^\pi$. Formally, we define the consolidation loss using real state sampling as

$$L^R_{consolid} = \mathbb{E}_{S \sim D_s} \left[ \sum_{A \in \mathcal{A}} \left( Q(S, A; \theta) - \hat{Q}(S, A; \theta^-) \right)^2 \right].$$

In practice, we sample states from the experience replay buffer $D$. We name our algorithm that uses real state sampling as memory-efficient DQN with real state sampling, denoted as *MeDQN(R)*. The algorithm description is shown in Algorithm 3 in the appendix. Similar to MeDQN(U), we also use the same data to train the $Q$ function for $E$ times and apply knowledge consolidation by adding a consolidation loss. The main difference is that the experience replay buffer used in MeDQN(R) is relatively large while the experience replay buffer in MeDQN(U) is extremely small (i.e., one mini-batch size). However, as we will show next, the experience replay buffer used in MeDQN(R) can still be significantly smaller than the experience replay buffer used in DQN.

## 5 Experiments

In this section, we first verify that knowledge consolidation helps mitigate the objective mismatch problem and reduce catastrophic forgetting. Next, we propose a strategy to better balance learning and remembering. We then show that our algorithms achieve similar or even better performance compared to DQN in both low-dimensional and high-dimensional tasks. Moreover, we verify that knowledge consolidation is the key to achieving memory efficiency and high performance with an ablation study. Finally, we show that knowledge consolidation makes our algorithms more robust to different buffer sizes. All experimental details, including hyper-parameter settings, are presented in the appendix.

### 5.1 The effectiveness of knowledge consolidation

To show that knowledge consolidation helps mitigate the objective mismatch problem, we first apply it to solve the task of approximating $\sin(\pi x)$, as presented in Section 2. Denote the neural network and the target neural network as $f(x; \theta)$ and $f(x; \theta^-)$, respectively. The true loss function is defined as $L_{true} = \mathbb{E}_{x \in [0,2]}[(f(x;\theta) - y)^2]$, where $y = \sin(\pi x)$. We first trained the network $f(x; \theta)$ in Stage 1 (i.e. $x \in [0, 1]$) without knowledge consolidation. We also maintained input bounds $[x_{LOW}, x_{HIGH}]$, similar to Algorithm 1. At the end of Stage 1, we set $\theta^- \leftarrow \theta$. At this moment, both $f(x; \theta)$ and $f(x; \theta^-)$ were good approximations of the true function for $x \in [0, 1]$; and $[x_{LOW}, x_{HIGH}] \approx [0, 1]$. Next, we continue to train $f(x; \theta)$ with $x \in [1, 2]$. To apply knowledge consolidation, we sampled $\hat{x}$ from input bounds $[x_{LOW}, x_{HIGH}]$ uniformly. Finally, we added the consolidation loss to the training loss and got:

$$L(\theta) = \mathbb{E}_{x \in [1,2]}[(f(x;\theta) - y)^2] + \mathbb{E}_{\hat{x} \in [x_{LOW}, x_{HIGH}]}[(f(\hat{x};\theta) - f(\hat{x};\theta^-))^2].$$

By adding the consolidation loss, $L$ became a good approximation of the true loss function $L_{true}$, which helps to preserve the knowledge learned in Stage 1 while learning new knowledge in Stage 2, as shown in Figure 1(b). Note that in the whole training process, we did not save previously observed samples $(x, y)$ explicitly. The knowledge of Stage 1 is stored in the target model $f(x; \theta^-)$ and then consolidated to $f(x; \theta)$.

Next, we show that combined with knowledge consolidation, MeDQN(U) is able to reduce the forgetting issue in single RL tasks even with a tiny replay buffer. We repeated the RL training process in Section 2 for MeDQN(U) and recorded its greedy action during training. As shown in Figure 2, while DQN(S) kept forgetting and relearning the optimal action, both DQN and MeDQN(U) were able to consistently choose the optimal action 1 as its greedy action, suffering much less for forgetting. Note that both MeDQN(U) and DQN(S) use a tiny replay buffer with size 32 while DQN uses a much larger replay buffer with size $10K$. This experiment demonstrated the forgetting issue in single RL tasks and the effectiveness of our method to mitigate forgetting. More training details in Mountain Car can be found in Section 5.3 and the appendix.

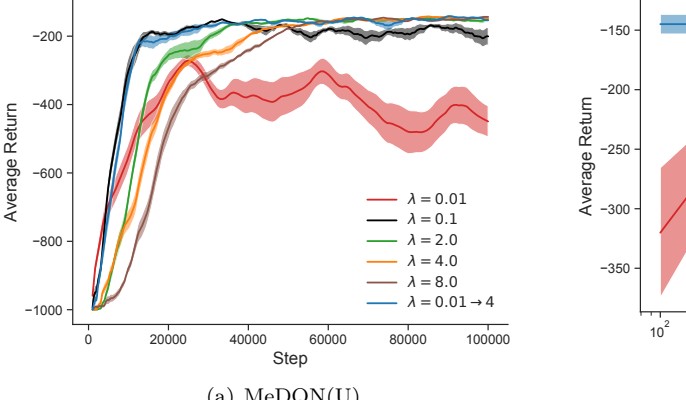
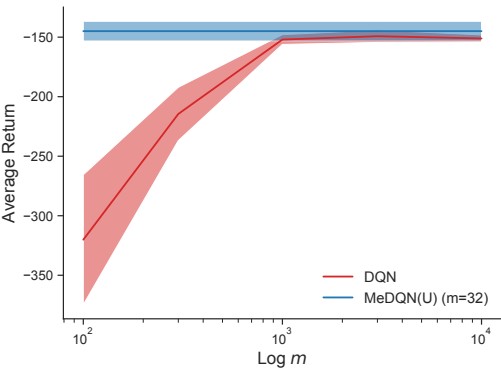

|(a) MeDQN(U)|(b) The influence of buffer size $m$.|

Figure 3: (a) Comparison of different strategies to balance learning and preservation for MeDQN(U) in MountainCar-v0. (b) A study of robustness to different buffer sizes $m$ in MountainCar-v0. For MeDQN(U), $m$ is fixed as the mini-batch size 32. DQN performs worse and worse as we decrease the buffer size while MeDQN(U) achieves high performance with an extremely small buffer. All results were averaged over 20 runs, with the shaded area representing two standard errors.

## 5.2 Balancing learning and remembering

In Section 4.2, we claimed that $\lambda$ could balance between learning new knowledge and preserving old knowledge. In this section, we used Mountain Car (Sutton & Barto, 2018) as a testbed to verify this claim. Specifically, a fixed $\lambda$ was chosen from $\{0.01, 0.1, 2, 4, 8\}$. The mini-batch size was 32. The experience replay buffer size in MeDQN(U) was 32. The update epoch $E = 4$, the target network update frequency $C_{target} = 100$, and the current network update frequency $C_{current} = 1$. We chose learning rate from $\{1e-2, 3e-3, 1e-3, 3e-4, 1e-4\}$ and reported the best results averaged over 20 runs for different $\lambda$, as shown in Figure 3(a).

When $\lambda$ is small (e.g., $\lambda = 0.1$ or $\lambda = 0.01$), giving a small weight to the consolidation loss, MeDQN(U) learns fast at the beginning. However, as training continues, the learning becomes slower and unstable; the performance drops. As we increase $\lambda$, although the initial learning is getting slower, the training process becomes more stable, resulting in higher performance. These phenomena align with our intuition. At first, not much knowledge is available for consolidation; learning new knowledge is more important. A small $\lambda$ lowers the weight of $L_{consolid}$ in the training loss, thus speeding up learning initially. As training continues, more knowledge is learned, and knowledge preservation is vital to performance. A small $\lambda$ fails to protect old knowledge, while a larger $\lambda$ helps consolidate knowledge more effectively, stabilizing the learning process.

Inspired by these results, we propose a new strategy to balance learning and preservation. Specifically, $\lambda$ is no longer fixed but linearly increased from a small value $\lambda_{start}$ to a large value $\lambda_{end}$. This mechanism encourages knowledge learning at the beginning and information retention in later training. We increased $\lambda$ from 0.01 to 4 linearly with respect to the training steps for this experiment. In this setting, we observed that learning is fast initially, and then the performance stays at a high level stably towards the end of training. Given the success of this linearly increasing strategy, we applied it in all of the following experiments for MeDQN.

## 5.3 Evaluation in low-dimensional tasks

We chose four tasks with low-dimensional inputs from Gym (Brockman et al., 2016) and PyGame Learning Environment (Tasfi, 2016): MountainCar-v0 (2), Acrobot-v1 (6), Catcher (4), and Pixelcopter (7), where numbers in parentheses are input state dimensions. For DQN, we used the same hyper-parameters and training settings as in Lan et al. (2020). The mini-batch size was 32. Note that the buffer size for DQN was 10,000 in all tasks. Moreover, we included *DQN(S)* as another baseline which used a tiny (32) experience replay buffer. The buffer size is the only difference between DQN and DQN(S). The experience replay buffer

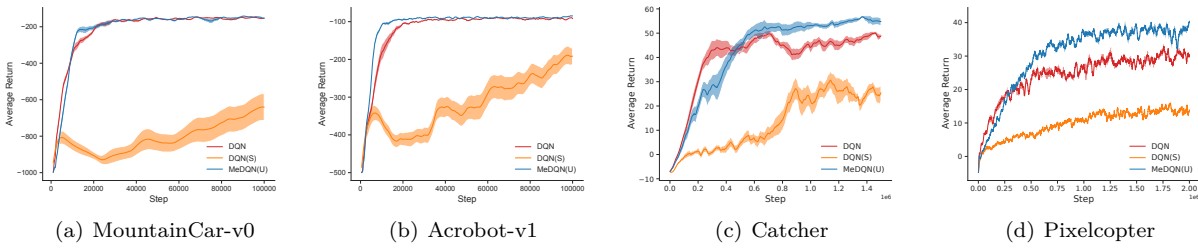

(a) MountainCar-v0          (b) Acrobot-v1          (c) Catcher          (d) Pixelcopter

Figure 4: Evaluation in low-dimensional tasks. The results for MountainCar-v0 and Acrobot-v1 were averaged over 20 runs. The results for Catcher and Pixelcopter were averaged over 10 runs. The shaded areas represent two standard errors. MeDQN(U) outperforms DQN in all four tasks, even though it uses an experience replay buffer with one mini-batch size.

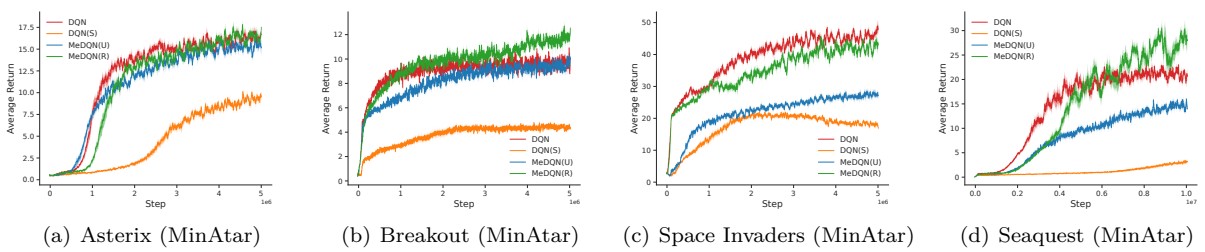

(a) Asterix (MinAtar)   (b) Breakout (MinAtar)   (c) Space Invaders (MinAtar)   (d) Seaquest (MinAtar)

Figure 5: Evaluation in high-dimensional tasks. All results were averaged over 10 runs, with the shaded areas representing two standard errors. MeDQN(R) is comparable with DQN even though it uses a much smaller experience replay buffer (10% of the replay buffer size in DQN).

size for MeDQN(U) was 32. We set $\lambda_{start} = 0.01$ for MeDQN(U) and tuned learning rate for all algorithms. Except for tuning the update epoch $E$, the current network update frequency $C_{current}$, and $\lambda_{end}$, we used the same hyper-parameters and training settings as DQN for MeDQN(U). More details are included in the appendix.

For MountainCar-v0 and Acrobot-v1, we report the best results averaged over 20 runs. For Catcher and Pixelcopter, we report the best results averaged over 10 runs. The learning curves of the best results for all algorithms are shown in Figure 4, where the shaded area represents two standard errors. As we can see from Figure 4, when using a small buffer, DQN(S) has higher instability and lower performance than DQN, which uses a large buffer. MeDQN(U) outperforms DQN in all four games. The results are very inspiring given that MeDQN(U) only uses a tiny experience replay buffer. We conclude that MeDQN(U) is much more memory-efficient while achieving similar or better performance compared to DQN in low-dimensional tasks. Given that MeDQN(U) already performs well while being very memory-efficient, we did not further test MeDQN(R) in these low-dimensional tasks.

## 5.4   Evaluation in high-dimensional tasks

To further evaluate our algorithms, we chose four tasks with high-dimensional image inputs from MinAtar (Young & Tian, 2019): Asterix ($10 \times 10 \times 4$), Seaquest ($10 \times 10 \times 10$), Breakout ($10 \times 10 \times 4$), and Space Invaders ($10 \times 10 \times 6$), where numbers in parentheses are input dimensions. For DQN, we reused the hyper-parameters and neural network setting in Young & Tian (2019). The mini-batch size was 32. For DQN, the buffer size was $100,000$ and the target network update frequency $C_{target} = 1,000$. For MeDQN(R), $C_{target} = 1,000$ and the buffer size was 10% of the buffer size in DQN, i.e. $10,000$. The experience replay buffer size for MeDQN(U) was 32. For MeDQN(U), a smaller $C_{target}$ was better, and we set $C_{target} = 300$. We set $\lambda_{start} = 0.01$ for MeDQN(R) and MeDQN(U) and tuned learning rate for all algorithms. We tuned

Table 1: An ablation study of knowledge consolidation for MeDQN(R) with different buffer sizes, tested in Seaquest (MinAtar). All final returns were averaged over 10 runs, shown with one standard error.

| Buffer Size | 1e5 | 1e4 | 1e3 |
|---|---|---|---|
| with consolidation | $25.82 \pm 1.69$ | $27.92 \pm 1.53$ | $17.22 \pm 0.90$ |
| w.o. consolidation | $20.10 \pm 0.71$ | $19.78 \pm 1.19$ | $16.04 \pm 0.60$ |

Table 2: A study of robustness to different buffer sizes, tested in Seaquest (MinAtar). All final returns were averaged over 10 runs, shown with one standard error.

| buffer size | 1e5 | 1e4 | 1e3 |
|---|---|---|---|
| MeDQN(R) | $25.82 \pm 1.69$ | $27.92 \pm 1.53$ | $17.22 \pm 0.90$ |
| DQN | $20.48 \pm 0.75$ | $21.38 \pm 0.80$ | $15.04 \pm 0.66$ |

the update epoch $E$, the current network update frequency $C_{current}$, and $\lambda_{end}$ for both MeDQN(R) and MeDQN(U), while reusing other hyper-parameters from DQN. More details are in the appendix.

The learning curves of best results are shown in Figure 5, averaging over 10 runs with the shaded areas representing two standard errors. Notice that with a small experience replay buffer, DQN(S) performs much worse than all other algorithms, including MeDQN(U), which also uses a small experience replay buffer of the same size as DQN(S). For tasks with a relatively lower input dimension, such as Asterix and Breakout, both MeDQN(R) and MeDQN(U) match up with or even outperform DQN. For Seaquest and Space Invaders which have larger input dimensions, MeDQN(R) is comparable with DQN even though it uses a much smaller experience replay buffer (10% of the replay buffer size in DQN). Meanwhile, the performance of MeDQN(U) is significantly lower than MeDQN(R), mainly due to different state sampling strategies. As discussed in Section 4.4, when the state space $\mathcal{S}$ is too large as the cases for Seaquest and Space Invaders, states generated with uniform state sampling can not cover visited states well enough, resulting in poor knowledge consolidation. In this circumstance, storing and sampling real states is usually a better approach. Overall, we conclude that MeDQN(R) is more memory-efficient than DQN while achieving comparably high performance and high sample efficiency.

### 5.5 An ablation study of knowledge consolidation

In this section, we present an ablation study of knowledge consolidation in Seaquest. By setting $\lambda_{start} = \lambda_{end} = 0$ in MeDQN(R), we removed the consolidation loss from the training loss and MeDQN(R) was reduced to DQN with $E$ updates for each mini-batch of transitions. *Except this, all other training settings were the same as MeDQN(R) with consolidation.* For example, we also tuned $E$ and $C_{current}$ for MeDQN(R) without consolidation. The averaged final returns with standard errors are presented in Table 1 which shows that MeDQN(R) performs better with consolidation than MeDQN(R) without consolidation under different buffer sizes. These results proved that knowledge consolidation is the key to the success of MeDQN(R); MeDQN(R) is not simply benefiting from choosing a higher $E$ or a larger $C_{current}$. It is the use of knowledge consolidation that allows MeDQN(R) to have a smaller experience replay buffer.

### 5.6 A study of robustness to different buffer sizes

Moreover, we study the performance of different algorithms with different buffer sizes on Seaquest (MinAtar), which has the largest input state space. In Table 2, we present the averaged returns at the end of training for MeDQN(R) and DQN with different buffer sizes. Clearly, MeDQN(R) is more robust than DQN to buffers at various scales. A similar study in a low-dimensional task (MountainCar-v0) is shown in Figure 3(b). Note that for MeDQN(U), $m$ is fixed as the mini-batch size 32. DQN performs worse and worse as we decrease the buffer size, while MeDQN(U) achieves high performance with a tiny buffer.

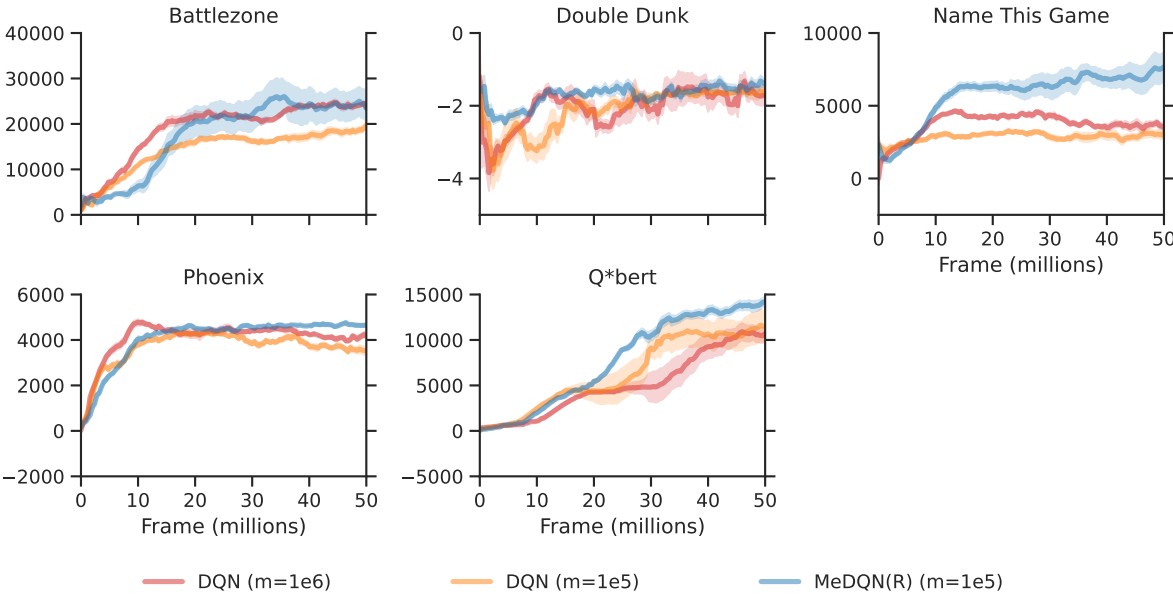

Figure 6: Return curves of various algorithms in five Atari tasks over 50 million training frames, with different buffer sizes $m$. Solid lines correspond to the median performance over 5 random seeds, and the shaded areas correspond to standard errors. Though using a much smaller buffer, MeDQN(R) ($m = 1e5$) outperforms DQN ($m = 1e6$) in three tasks significantly (i.e., Name This Game, Phoenix, and Qbert) and matches DQN's performance in the other two tasks.

### 5.7 Additional results in Atari games

To further evaluate our algorithm, we conducted experiments in Atari games (Bellemare et al., 2013). Specifically, we selected five representative games recommended by Aitchison et al. (2022), including Battlezone, Double Dunk, Name This Game, Phoenix, and Qbert. The input dimensions for all five games are $84 \times 84$.

We compared our algorithm MeDQN(R) and DQN with different buffer sizes. Specifically, we used the implementation of DQN [3] in Tianshou (Weng et al., 2022) and implemented MeDQN(R) based on Tianshou's DQN. For DQN, we used the default hyper-parameters of Tianshou's DQN. For MeDQN, $\lambda_{start} = 0.01$; $\lambda_{end}$ was chosen from $\{2, 4\}$; $E = 1$. We chose $C_{current}$ in $\{20, 40\}$ while the default $C_{current} = 10$ in Tianshou's DQN. Except those hyper-parameters, other hyper-parameters of MeDQN are the same as the default hyper-parameters of Tianshou's DQN.

We train all algorithms for 50M frames (i.e., 12.5M steps) and summarize results across over 5 random seeds in Figure 6. The solid lines correspond to the median performance over 5 random seeds, while the shaded areas represent standard errors. Though using a much smaller buffer, MeDQN(R) ($m = 1e5$) outperforms DQN ($m = 1e6$) in three tasks significantly (i.e., Name This Game, Phoenix, and Qbert) and matches DQN's performance in the other two tasks. Overall, the memory usage of a replay buffer is reduced from 7GB to 0.7GB without hurting the agent's performance, confirming our previous claim that MeDQN(R) is more memory-efficient than DQN while achieving comparably high performance.

In the appendix, we report the actual memory usage and the training wall-clock time of training in four MinAtar games and five Atari games. Generally, MeDQN(U) and MeDQN(R) are more computation-efficient since they require less frequent updates due to less forgetting. Considering all above results, we suggest using MeDQN(U) for low-dimensional value-based control tasks and MeDQN(R) for high-dimensional value-based control tasks.

---

[3]https://github.com/thu-ml/tianshou/blob/master/examples/atari/atari_dqn.py

## 6  Conclusion and future work

In this work, we proposed two memory-efficient algorithms based on DQN. Our algorithms can find a good trade-off between learning new knowledge and preserving old knowledge. By conducting rigorous experiments, we showed that a large experience replay buffer could be successfully replaced by (real or uniform) state sampling, which costs less memory while achieving good performance and high sample efficiency.

There are many open directions for future work. For example, a pre-trained encoder can be used to reduce high-dimensional inputs to low-dimensional inputs (Hayes et al., 2019; 2020; Chen et al., 2021), boosting the performance of MeDQN(U) on high-dimensional tasks. Combining classic memory-saving methods (Schlegel et al., 2017) with our knowledge consolidation approach may further reduce memory requirement and improve sample efficiency. Generative models (e.g., VAE (Kingma & Welling, 2013) or GAN (Goodfellow et al., 2014)) could be applied to approximate $d^\pi$. The challenge for this approach would be to generate realistic pseudo-states to improve knowledge consolidation and reduce forgetting. It is also worth considering the usage of various sampling methods for the knowledge consolidation loss, such as stochastic gradient Langevin dynamics methods (Pan et al., 2022b; 2020). A combination of uniform state sampling and real state sampling might further improve the agent's performance. Finally, extending our ideas to policy gradient methods would also be interesting.

### Acknowledgments

We gratefully acknowledge funding from the Canada CIFAR AI Chairs program, the Reinforcement Learning and Artificial Intelligence (RLAI) laboratory, the Alberta Machine Intelligence Institute (Amii), and the Natural Sciences and Engineering Research Council (NSERC) of Canada.

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

## A The pseudocodes of DQN and MeDQN(R)

---

**Algorithm 2** Deep Q-learning with experience replay (DQN)

---

1: Initialize an experience replay buffer $D$
2: Initialize action-value function $Q$ with random weights $\theta$
3: Initialize target action-value function $\hat{Q}$ with random weights $\theta^- = \theta$
4: Observe initial state $s$
5: **while** agent is interacting with the environment **do**
6:     Choose action $a$ by $\epsilon$-greedy based on $Q$
7:     Take action $a$, observe $r$, $s'$
8:     Store transition $(s, a, r, s')$ in $D$ and update state $s \leftarrow s'$
9:     **for** every $C_{current}$ steps **do**
10:       Sample a random mini-batch of transitions $(s_B, a_B, r_B, s'_B)$ from $D$
11:       Get update target: $Y_B \leftarrow r_B + \gamma \max_{a' \in \mathcal{A}} \hat{Q}(s'_B, a'; \theta^-)$
12:       Compute DQN loss: $L_{DQN} = (Y_B - Q(s_B, a_B; \theta))^2$
13:       Perform a gradient descent step on $L_{DQN}$ with respect to $\theta$
14:     **end for**
15:     Reset $\hat{Q} = Q$ for every $C_{target}$ steps
16: **end while**

---

**Algorithm 3** Memory-efficient DQN with real state sampling

---

1: Initialize an experience replay buffer $D$
2: Initialize action-value function $Q$ with random weights $\theta$
3: Initialize target action-value function $\hat{Q}$ with random weights $\theta^- = \theta$
4: Observe initial state $s$
5: **while** agent is interacting with the environment **do**
6:     Choose action $a$ by $\epsilon$-greedy based on $Q$
7:     Take action $a$, observe $r$, $s'$
8:     Store transition $(s, a, r, s')$ in $D$ and update state $s \leftarrow s'$
9:     **for** every $C_{current}$ steps **do**
10:       Sample a random mini-batch of transitions $(s_B, a_B, r_B, s'_B)$ from $D$
11:       **for** $i = 1$ **to** $E$ **do**
12:         Compute DQN loss $L_{DQN}$
13:         Sample a random mini-batch of states $s_{B'}$ from $D$
14:         Compute consolidation loss $L^R_{consolid}$ given $s_{B'}$
15:         Compute the final training loss: $L = L_{DQN} + \lambda L^R_{consolid}$
16:         Perform a gradient descent step on $L$ with respect to $\theta$
17:       **end for**
18:     **end for**
19:     Reset $\hat{Q} = Q$ for every $C_{target}$ steps
20: **end while**

---

## B The relation between knowledge consolidation and trust region methods

Trust region methods such as TRPO (Schulman et al., 2015) and PPO (Schulman et al., 2017) use KL divergence to prevent large policy updates. In their implementations (Achiam, 2018), the same states are used for TD learning and policy constraints, while we sample different states in our methods. Thus, policy constraints in trust region methods are close to knowledge consolidation with real state sampling where the sampled states for policy constraints are the same as the states for TD learning. Ideally, to reduce forgetting, we only want to update the Q values and policy of states used in TD learning; the rest of Q values and policy should remain the same. Using the same states for TD learning and policy constraint cannot achieve this.

# C    Implementation details

## C.1    Supervised learning: approximating a sine function

The goal of this task is to approximate a sine function $y = \sin(\pi x)$, where $x \in [0, 2]$. To get non-IID input data, we consider two-stage training. In Stage 1, we generated training samples $(x, y)$, where $x \in [0, 1]$ and $y = \sin(\pi x)$. For Stage 2, $x \in [1, 2]$ and $y = \sin(\pi x)$. The neural network was a multi-layer perceptron with hidden layers $[32, 32]$ and ReLU activation functions. We used Adam optimizer with learning rate 0.01. The mini-batch size was 32. For each stage, we trained the network with $1,000$ mini-batch updates.

## C.2    Low-dimensional tasks

For MountainCar-v0 and Acrobot-v1, the depicted return in Figure 4 was averaged over the last 20 episodes; the neural network was a multi-layer perceptron with hidden layers $[32, 32]$; the best learning rate was selected from $\{1e-2, 3e-3, 1e-3, 3e-4, 1e-4\}$ with grid search; Adam was used to optimize network parameters; all algorithms were trained for $1e5$ steps.

For Catcher and Pixelcopter, the depicted return in Figure 4 was averaged over the last 100 episodes; the neural network was a multi-layer perceptron with hidden layers $[64, 64]$; the best learning rate was selected from $\{1e-3, 3e-4, 1e-4, 3e-5, 1e-5\}$ with grid search; RMSprop was used to optimize network parameters. In Catcher, algorithms were trained for $1.5e6$ steps; in Pixelcopter, algorithms were trained for $2e6$ steps.

All curves were smoothed using an exponential average. The discount factor was 0.99. The mini-batch size was 32. For first $1,000$ exploration steps, we only collected transitions without learning. $\epsilon$-greedy was applied as the exploration strategy with $\epsilon$ decreasing linearly from 1.0 to 0.01 in $1,000$ steps. After $1,000$ steps, $\epsilon$ was fixed to 0.01. For MeDQN, $\lambda_{start} = 0.01$; $\lambda_{end}$ was chosen from $\{1, 2, 4\}$; $E$ was selected from $\{1, 2, 4\}$. We chose $C_{current}$ in $\{1, 2, 4, 8\}$. Other hyper-parameter choices are presented in Table 3–6.

Table 3: The hyper-parameters of different algorithms for **MountainCar-v0**, used in Figure 4(a).

| Hyper-parameter | DQN | DQN(S) | MeDQN(U) |
|---|---|---|---|
| learning rate | 1e-2 | 1e-3 | 1e-3 |
| experience replay buffer size | 1e4 | 32 | 32 |
| $E$ | / | / | 4 |
| $\lambda_{end}$ | / | / | 4 |
| $C_{target}$ | 100 | 100 | 100 |
| $C_{current}$ | 8 | 1 | 1 |

Table 4: The hyper-parameters of different algorithms for **Acrobot-v1**, used in Figure 4(b).

| Hyper-parameter | DQN | DQN(S) | MeDQN(U) |
|---|---|---|---|
| learning rate | 1e-3 | 3e-4 | 3e-4 |
| experience replay buffer size | 1e4 | 32 | 32 |
| $E$ | / | / | 1 |
| $\lambda_{end}$ | / | / | 2 |
| $C_{target}$ | 100 | 100 | 100 |
| $C_{current}$ | 1 | 1 | 1 |

## C.3    High-dimensional tasks: MinAtar

The depicted return in Figure 5 was averaged over the last 500 episodes, and the curves were smoothed using an exponential average. In Seaquest (MinAtar), all algorithms were trained for $1e7$ steps. Except that, all algorithms were trained for $5e6$ steps in other tasks. The discount factor was 0.99. The mini-batch size was 32.

Table 5: The hyper-parameters of different algorithms for **Catcher**, used in Figure 4(c).

| Hyper-parameter | DQN | DQN(S) | MeDQN(U) |
|---|---|---|---|
| learning rate | 1e-4 | 1e-5 | 3e-5 |
| experience replay buffer size | 1e4 | 32 | 32 |
| $E$ | / | / | 4 |
| $\lambda_{end}$ | / | / | 4 |
| $C_{target}$ | 200 | 200 | 200 |
| $C_{current}$ | 1 | 1 | 2 |

Table 6: The hyper-parameters of different algorithms for **Pixelcopter**, used in Figure 4(d).

| Hyper-parameter | DQN | DQN(S) | MeDQN(U) |
|---|---|---|---|
| learning rate | 3e-5 | 1e-5 | 3e-4 |
| experience replay buffer size | 1e4 | 32 | 32 |
| $E$ | / | / | 1 |
| $\lambda_{end}$ | / | / | 1 |
| $C_{target}$ | 200 | 200 | 200 |
| $C_{current}$ | 1 | 1 | 8 |

For the first $5,000$ exploration steps, we only collected transitions without learning. $\epsilon$-greedy was applied as the exploration strategy with $\epsilon$ decreasing linearly from 1.0 to 0.1 in $5,000$ steps. After $5,000$ steps, $\epsilon$ was fixed to 0.1. Following Young & Tian (2019), we used smooth L1 loss in PyTorch and centered RMSprop optimizer with $\alpha = 0.95$ and $\epsilon = 0.01$. The best learning rate was chosen from $\{3e-3, 1e-3, 3e-4, 1e-4, 3e-5\}$ with grid search. We also reused the settings of neural networks. For MeDQN, $\lambda_{start} = 0.01$; $\lambda_{end}$ was chosen from $\{2, 4\}$; $E$ was selected from $\{1, 2\}$. We chose $C_{current}$ in $\{4, 8, 16, 32\}$. Other hyper-parameter choices are presented in Table 7–10.

Table 7: The hyper-parameters of different algorithms for **Asterix (MinAtar)**, used in Figure 5(a).

| Hyper-parameter | DQN | DQN(S) | MeDQN(U) | MeDQN(R) |
|---|---|---|---|---|
| learning rate | 1e-4 | 3e-5 | 1e-3 | 3e-4 |
| experience replay buffer size | 1e5 | 32 | 32 | 1e4 |
| $E$ | / | / | 2 | 2 |
| $\lambda_{end}$ | / | / | 2 | 2 |
| $C_{target}$ | 1000 | 1000 | 300 | 1000 |
| $C_{current}$ | 1 | 1 | 16 | 8 |

## D   Computation and memory usage

For our experiments, we used computation resources (about 7 CPU core years and 0.25 GPU core year) from the Digital Research Alliance of Canada.

Table 11 presents a complete comparison of buffer sizes for different algorithms. Table 12 presents a comparison of actual memory usage of training different algorithms for Seaquest (MinAtar). Table 13 presents a comparison of actual memory usage of training different algorithms in Atari games. Table 14 presents a comparison of actual wall-clock time of training different algorithms for MinAtar games. Table 15 presents a comparison of training speed of different algorithms for Atari games.

Table 8: The hyper-parameters of different algorithms for **Breakout (MinAtar)**, used in Figure 5(b).

| Hyper-parameter | DQN | DQN(S) | MeDQN(U) | MeDQN(R) |
|---|---|---|---|---|
| learning rate | 1e-3 | 3e-5 | 3e-3 | 3e-4 |
| experience replay buffer size | 1e5 | 32 | 32 | 1e4 |
| $E$ | / | / | 1 | 2 |
| $\lambda_{end}$ | / | / | 2 | 4 |
| $C_{target}$ | 1000 | 1000 | 300 | 1000 |
| $C_{current}$ | 1 | 1 | 8 | 4 |

Table 9: The hyper-parameters of different algorithms for **Space Invaders (MinAtar)**, used in Figure 5(c).

| Hyper-parameter | DQN | DQN(S) | MeDQN(U) | MeDQN(R) |
|---|---|---|---|---|
| learning rate | 3e-4 | 3e-5 | 1e-3 | 3e-4 |
| experience replay buffer size | 1e5 | 32 | 32 | 1e4 |
| $E$ | / | / | 1 | 2 |
| $\lambda_{end}$ | / | / | 4 | 2 |
| $C_{target}$ | 1000 | 1000 | 300 | 1000 |
| $C_{current}$ | 1 | 1 | 32 | 4 |

Table 10: The hyper-parameters of different algorithms for **Seaquest (MinAtar)**, used in Figure 5(d).

| Hyper-parameter | DQN | DQN(S) | MeDQN(U) | MeDQN(R) |
|---|---|---|---|---|
| learning rate | 1e-4 | 3e-5 | 3e-3 | 3e-4 |
| experience replay buffer size | 1e5 | 32 | 32 | 1e4 |
| $E$ | / | / | 1 | 1 |
| $\lambda_{end}$ | / | / | 2 | 4 |
| $C_{target}$ | 1000 | 1000 | 300 | 1000 |
| $C_{current}$ | 1 | 1 | 32 | 4 |

Table 11: Comparison of buffer sizes in low- and high-dimensional tasks: MinAtar. Compared to DQN which uses a large experience replay buffer, MeDQN is much more memory-efficient while achieving comparable performance.

| Task | DQN | DQN(S) | MeDQN(U) | MeDQN(R) |
|---|---|---|---|---|
| low-dimensional | 10K | 32 | 32 | / |
| high-dimensional: MinAtar | 100K | 32 | 32 | 10K |
| high-dimensional: Atari | 1M | / | / | 100K |

Table 12: Comparison of actual memory usage of different algorithms in Seaquest (MinAtar). Here, the actual memory usage refers to the resident set size (including experience replay buffer, neural networks, etc.), obtained with Python package *psutil*.

|  | **DQN** | **DQN(S)** | **MeDQN(U)** | **MeDQN(R)** |
|---|---|---|---|---|
| Memory (MB) | 1400 | 240 | 250 | 280 |

Table 13: Comparison of actual memory usage of different algorithms in Atari games. Here, the actual memory usage refers to the resident set size (including experience replay buffer, neural networks, etc.), obtained with Python package *psutil*.

|  | **DQN** $(m = 1e6)$ | **DQN** $(m = 1e5)$ | **MeDQN(R)** $(m = 1e5)$ |
|---|---|---|---|
| Memory (GB) | 8.9 | 2.9 | 3.1 |

## E   Why uniform state sampling works

The intuition behind uniform state sampling is that sometimes we do not necessarily need real states (i.e. states sampled from $d^\pi$) to achieve good knowledge consolidation. Although randomly uniformly generated states may not be meaningful, they are enough to induce perfect knowledge consolidation (i.e. $L_{consolid} = 0$) in some cases. For example, considering linear function approximations that $Q(s, a; \theta) = x^\top \theta$ and $Q(s, a; \theta^-) = x^\top \theta^-$, where $\theta \in \mathbb{R}^n$, $\theta^- \in \mathbb{R}^n$, and $x = (s, a) \in \mathbb{R}^n$. To achieve perfect knowledge consolidation ($L_{consolid} = 0$), it is required to find $\theta = \theta^-$ by minimizing $L_{consolid}$. Let $\{x_1, x_2, \ldots, x_n\}$ be $n$ points randomly uniformly sampled from $\mathcal{S} \times \mathcal{A}$ and set $y_i = x_i^\top \theta^-$ for every $i$. Denote $X = [x_1; \cdots; x_n]$ and $Y = [y_1, \cdots, y_n]^\top$. The problem can be reformulated as finding $\theta$ such that $X^\top \theta = Y$. It is known that the random matrix $X$ is full rank with a high probability (Cooper, 2000). In this case, we can get the optimal $\theta$ with $\theta = (X^\top)^{-1} Y$ and thus achieve perfect knowledge consolidation ($L_{consolid} = 0$). Note that $\{x_1, x_2, \ldots, x_n\}$ are all randomly generated; they are not necessarily to be real state-action pairs sampled from real trajectories.

Table 14: The wall-clock training time (in hours) of different algorithms on four MinAtar games for results shown in Figure 5. MeDQN(U) and MeDQN(R) take less time to train and are more computation-efficient than DQN.

| Task | DQN | DQN(S) | MeDQN(U) | MeDQN(R) |
|---|---|---|---|---|
| Asterix (MinAtar) | 3.83 | 3.68 | 1.17 | 1.78 |
| Breakout (MinAtar) | 3.83 | 3.68 | 1.17 | 2.67 |
| Space Invaders (MinAtar) | 4.08 | 3.83 | 0.75 | 3.33 |
| Seaquest (MinAtar) | 9.91 | 8.25 | 1.33 | 4.67 |

Table 15: The training speed (step/s) of different algorithms in five Atari games for results shown in Figure 6. MeDQN(R) is faster to train and more computation-efficient than DQN.

| Task | DQN $(m = 1e6)$ | DQN $(m = 1e5)$ | MeDQN(R) $(m = 1e5)$ |
|---|---|---|---|
| Battlezone (Atari) | 230 | 230 | 330 |
| Double Dunk (Atari) | 230 | 230 | 350 |
| Name This Game (Atari) | 240 | 240 | 350 |
| Phoenix (Atari) | 270 | 270 | 320 |
| Qbert (Atari) | 240 | 240 | 280 |

