# OpenReview forum: "Memory-efficient Reinforcement Learning with Value-based Knowledge Consolidation"
_TMLR — Accepted by TMLR_

### Review · Reviewer_uL1k · 2023-02-27

**Summary Of Contributions:**

Utilizing a replay buffer is one of the important practical techniques in deep off-policy RL. However, keeping the replay buffer increases the memory requirement. This paper proposes a memory-efficient method for off-policy RL algorithms. The main idea is using a small replay buffer and adding consistency regularization (called knowledge consolidation). By adding knowledge consolidation, the authors prevent the catastrophic forgetting issue. For knowledge consolidation, the authors propose two sampling schemes: uniform sampling which generates random samples from uniform distribution, and real state sampling utilizes previous states in the buffer. The authors showed that the proposed method achieves good performances with small replay buffer on simple OpenAI Gym envs and MiniAtar envs.

**Audience:**

Yes

**Broader Impact Concerns:**

There are no particular concerns on the ethical implications.

**Claims And Evidence:**

No

**Requested Changes:**

* Evaluation on more complicated domains: in order to show the (significant) benefit of this simple idea, more evaluations on complicated domains are required b/c MiniAtari and OpenAI Gym are too simple. I think testing the ideas on domains where a big replay buffer is required (e.g., more high-resolution images-- https://aihabitat.org/-- or requiring many samples due to complexity--humanoid) can make this work more convincing.

* Extension to continuous action space.

**Strengths And Weaknesses:**

Strengths

* Simplicity of the proposed method: the proposed idea is easy to implement (but at the same time, novelty is a bit limited).

* Clear writing

Weaknesses

* Novelty: The core part of this work (i.e., knowledge consolidation) is not a very new idea and the uniform sampling method doesn't sound like a general solution. I'm not sure if this proposed idea can be working on more complex domains with realistic images (https://aihabitat.org/) or complex dynamics (humanoid).

* Current results on simple domains are not very convincing (see requested changes).

---

> ### Author Response · Authors · 2023-03-15
> **We thank you for the detailed feedback, suggestions, and comments.**
>
> We thank you for the detailed feedback, suggestions, and comments. In the following, we address your concerns one by one.
> 1. **The core part of this work (i.e., knowledge distillation) is not a very new idea.** We agree that knowledge distillation is not novel. Proposed by Hinton et al. (2014), knowledge distillation is a useful tool, widely applied in computer vision [1], natural language processing [2], and reinforcement learning [3]. The tool itself is not novel, but how we use it is novel. Our contribution is showing that knowledge distillation can be combined with DQN to effectively reduce catastrophic forgetting in single RL tasks.
> 2. **Extension to continuous action space.** As mentioned by Review xdVG, the continuous action space setting can be more complex in many ways. We leave the investigation of this extension for future work. Furthermore, in this work, we already showed that 1) catastrophic forgetting exists in single RL tasks under memory constraints and 2) our methods can effectively reduce forgetting and maintain high performance under memory constraints, which we believe is a substantial contribution.
> 3. **Evaluation on more complicated domains.** Please check our general response.
>
> [1] Shin, Hanul, et al. "Continual learning with deep generative replay." Advances in neural information processing systems 30 (2017).
> [2] Tang, Raphael, et al. "Distilling task-specific knowledge from bert into simple neural networks." arXiv preprint arXiv:1903.12136 (2019).
> [3] Schwarz, Jonathan, et al. "Progress & compress: A scalable framework for continual learning." International conference on machine learning. PMLR, 2018.

---

> ### Author Response · Authors · 2023-03-25
> **Official Comment by Authors**
>
> Dear Reviewer uL1k,
>
> Thank you so much for taking the time to review our paper. We sincerely appreciate your detailed feedback and appraisal of our work, and have carefully addressed your comments in our rebuttal response. As the author-reviewer discussion period is half-way through, we would be grateful if you could acknowledge receipt of our responses and let us know if they address your concerns. We are eager to engage in any further discussions if needed.

---

> > ### Comment · Reviewer_uL1k · 2023-03-27
> > **response**
> >
> > Thank you for your response. I think this work tackles a very important problem (reducing the size of the replay buffer for off-policy RL) in practice. However, my biggest concern is that the proposed method is too synthetic, and tested environments are too simple. Even though the authors justified their this in the rebuttal, evaluations on more complex environments are required. Here, complex environments imply some environments requiring many interactions with environments (thus big replay buffer is required). For example, even for atari, I wonder if this method is working on hard exploration games like Montezuma's revenge. For locomotion tasks, evaluation on some environments like humanoid can be much more interesting because it consists of high-dimensional state and action spaces increasing the number of samples for training. My biggest question is how many researchers will be convinced by the proposed approach and adapt this to their problems where a large replay buffer is a big bottleneck.

---

> > > ### Author Response · Authors · 2023-03-27
> > > **Further response**
> > >
> > > Thank you for taking time to share your feedback on our work. We appreciate your comments and would like to address your concerns.
> > >
> > > Firstly, we want to clarify that Atari games with an input state size of $84 \times 84$, which is approximately 7K, are widely considered as challenging high-dimensional tasks. As such, they are often used as benchmarks for many DQN-based algorithms [1-6]. As presented in appendix D & E, we compared our algorithm with DQN in 5 Atari games, including Q*bert, one of the hard exploration tasks [7]. Using our algorithm, we can reduce the memory size of a replay buffer from 7GB to 0.7GB without hurting the agent's performance.
> > >
> > > Regarding Montezuma's revenge, we acknowledge that DQN and many DQN variants fail in this game [8]. As value-based methods, our algorithm and DQN are not applicable in continuous control tasks, such as Humanoid. We agree with Reviewer xdVG that working on continuous control tasks requires a different set of algorithms, i.e. policy gradient methods. We make no claims and have no reasons to believe that our approach will easily extend to this setting. It is unreasonable to invent a new algorithm that is completely different.
> > >
> > > We admit that our work is limited to DQN, and we only claim that our method is competitive with DQN while using substantially less memory. We did not make a claim on tasks where DQN does not perform well (e.g. Montezuma's revenge) or is not applicable (e.g. continuous control tasks). We could further clarify the scope of our work to make it more specific to value-based methods only.
> > >
> > > We understand that our work may not be impactful enough to convince all researchers to adapt it to their problems. However, we do believe that our work is interesting enough that some researchers could learn and benefit from it. In fact, according to TMLR's [acceptance criteria](https://jmlr.org/tmlr/acceptance-criteria.html), it is recommended to focus on the notion of "interest" rather than impact, which we believe our work has satisfied.
> > >
> > > Once again, we appreciate your feedback and are always striving to improve our methods and addressing any concerns or questions. Thank you.
> > >
> > > [1] Mnih, V., Kavukcuoglu, K., Silver, D., Antonoglou, A. G. I., Wierstra, D., & Riedmiller, M. Playing Atari with Deep Reinforcement Learning.
> > > [2] Van Hasselt, H., Guez, A., & Silver, D. (2016, March). Deep reinforcement learning with double q-learning. In Proceedings of the AAAI conference on artificial intelligence (Vol. 30, No. 1).
> > > [3] Bellemare, M. G., Dabney, W., & Munos, R. (2017, July). A distributional perspective on reinforcement learning. In International conference on machine learning (pp. 449-458). PMLR.
> > > [4] Mnih, V., Kavukcuoglu, K., Silver, D., Rusu, A. A., Veness, J., Bellemare, M. G., ... & Hassabis, D. (2015). Human-level control through deep reinforcement learning. nature, 518(7540), 529-533.
> > > [5] Hessel, M., Modayil, J., Van Hasselt, H., Schaul, T., Ostrovski, G., Dabney, W., ... & Silver, D. (2018, April). Rainbow: Combining improvements in deep reinforcement learning. In Proceedings of the AAAI conference on artificial intelligence (Vol. 32, No. 1).
> > > [6] Li, Z., Li, Y., Zhang, Y., Zhang, T., & Luo, Z. Q. HyperDQN: A Randomized Exploration Method for Deep Reinforcement Learning. In International Conference on Learning Representations.
> > > [7] Bellemare, M., Srinivasan, S., Ostrovski, G., Schaul, T., Saxton, D., & Munos, R. (2016). Unifying count-based exploration and intrinsic motivation. Advances in neural information processing systems, 29.
> > > [8] Quan, J., & Ostrovski, G. (2020). DQN Zoo: Reference implementations of DQN-based agents.

---

### Review · Reviewer_qCbE · 2023-03-03

**Summary Of Contributions:**

The paper proposes a technique to improve learning in the context of reinforcement learning, in particular to allow using a smaller replay buffer without losing (much) in performance. The algorithm is based on the deep Q-network algorithm. The algorithm introduces a "consolidation loss" that reduces forgetting from the target Q-network to the current Q-network. The algorithm is compared to the baseline without the suggested improvement and show that it achieves comparable or better performance with smaller experience replay buffers.

**Audience:**

Yes

**Broader Impact Concerns:**

No concern

**Claims And Evidence:**

Yes

**Requested Changes:**

- (suggestion to strengthen work) The positioning of the work with respect to the related work could be improved: the difference with methods such as EWC (Kirkpatrick et al., 2017) and SI (Zenke et al., 2017) are mentioned. In the related work, it is however not fully clear whether there are other algorithms that regularize the function directly in the context of (continual) supervised learning. If not, why is the contribution focused on the context of RL only? Is yes, could it be viewed as an extension to RL of existing techniques?
- (suggestion to strengthen work) Rewrite more concisely Sections 4.3 and 4.4. It seems that the overall idea could be summarized in just a few lines for each sections. It is a bit unclear why some of the elements are explained in great lengths, while also introducing some notations in ways that might bring more confusion than values to the paper. For instance, the definition of S_{LOW} and s_{HIGH} is not very clear and probably not needed in my opinion. Some technical details could be put in the appendix and the most important parts could be summarized in the paper.
- (suggestion to strengthen work) Concerning the environments, it is tested on 4 low dim environments and 4 high-dim ones (Atari). Overall 8 environments is already good according to me (more is even better, e.g. 6-8Atari games would already give a broader view). But given the compute requirements, one alternative (or something in addition) to having more environments is to clarify why/how these environments were selected. Were they at random in the set of Atari games or chosen for some specific reasons? This can be important to judge whether we can expect that the same performance would likely be obtained on other environments or whether we can expect that the improved performance is limited to specific types of environments.

**Strengths And Weaknesses:**

Strengths
- The paper is overall well written
- The method is interesting and simple to implement
- The experimental evaluation matches the claims of the paper

Weaknesses
- Some parts could have been written more concisely to focus on the core aspect of the contribution while potentially leaving some parts for the appendix (e.g. It seems possible to summarize more concisely the main ideas from sections 4.3 and 4.4).
- The experiments are sufficient in my opinion but one could always argue about more environments

---

> ### Author Response · Authors · 2023-03-15
> **We thank you for the detailed feedback, suggestions, and comments.**
>
> We thank you for the detailed feedback, suggestions, and comments. In the following, we address your concerns one by one.
> 1. **Are the proposed methods an extension of existing techniques to RL?** As mentioned in the related work, our method is inspired by generative replay methods in continual supervised learning. Generative replay methods exploit a dual memory system consisting of a student and a teacher network. The current training samples from a data buffer are combined with pseudo samples generated from the teacher network and then used to train the student network with knowledge distillation. In our work, the target Q network regularizes the current Q network, playing a similar (but not the same) role as the teacher network. For example, we do not use the target Q network to generate pseudo samples. We have updated the related work to directly point out other methods that regularize the function directly in the context of continual supervised learning.
> 2. **Rewrite Sections 4.3 and 4.4 more concisely.** We have rewritten these two sections more concisely. However, we do want to emphasize the importance of $s_{LOW}$ and $s_{HIGH}$ as they are essential elements in our algorithm MeDQN(U) (see Line 2,7,14 in Algorithm 1 for details). We added more explanations to clarify their definitions in Section 4.3. Please let us know if you need further clarification.
> 3. **Why and how are these MinAtar games selected?** Please check our general response.

---

> > ### Comment · Reviewer_qCbE · 2023-03-20
> > **Clear replies, one more question**
> >
> > Thank you for the clear answer and modifications to the paper.
> >
> > Concerning the "Uniform state sampling", I'm wondering how this can be efficient in high-dimensional states where structure is key. The uniform sampling states process would likely select frames that look like white noise in many cases as explained in the paper. However, the results are still better than DQN in the MinAtar environments. Can you comment on whether you think it is still significantly better in these cases or whether this method is *only* useful for low-dimensional environments? There is already a discussion in the paper but this point is a bit vague with comments such as "storing and sampling real states is usually a better approach".

---

> > > ### Author Response · Authors · 2023-03-20
> > > **Reply to question related to uniform state sampling**
> > >
> > > To rewrite Sections 4.3 more concisely, we moved a short discussion about uniform state sampling to Appendix E. Please check.
> > > The intuition is that with enough sampled states, we can achieve good enough knowledge consolidation, even with "white noise" states. For a low-dimensional state space, a small number of states are enough to cover the whole state space well, achieving good consolidation. However, as the dimension of the state space increases, we may need to sample exponentially more pseudo-states to cover the whole state space and compute the consolidation loss, which is computation expensive. This limits the application of uniform state sampling in high-dimensional tasks.
> > > However, even in the case of high-dimensional state spaces, uniform state sampling still helps as we've shown in MinAtar experiments. It is just not helpful enough to beat DQN with a large replay buffer.
> > >
> > > As mentioned in the future work (Section 6), we may incorporate a pre-trained encoder to map high-dimensional observations to low-dimensional state features (Hayes et al., 2019; 2020; Chen et al., 2021), since the intrinsic dimension of observations might actually be low. We could then apply uniform state sampling over the low-dimensional compressed state features which may further boost the performance of MeDQN(U) in high-dimensional tasks.
> > >
> > > We hope our reply is helpful for you. Please let us know if you need further clarification.

---

### Review · Reviewer_xdVG · 2023-03-07

**Summary Of Contributions:**

- Authors point out the problem that a large buffer can use up a lot of memory, making it difficult to deploy DRL to portable devices.
- Authors propose a new technique to reduce forgetting in the training process to allow better performance with a smaller buffer. The core idea is to have a consolidation loss, which essentially moves current Q estimates towards target Q network estimates. Two algorithmic variants are proposed, MeDQN(U) uses a very small buffer and sample uniformly from a bounded state space, MeDQN(R) uses a larger buffer (but still much smaller than DQN baseline) and sample from the buffer.
- Extensive experiments are ablations are provided for a number of tasks. Authors show that the proposed method can achieve much stronger performance compared to baseline when buffer size is small.

**Audience:**

Yes

**Claims And Evidence:**

Yes

**Requested Changes:**

- Add comparison to related techniques that might help tackle forgetting, test them with smaller buffer sizes to see how they compare to your proposed method. Just comparing to the baseline does not tell us how good the proposed method is against other methods in the literature.
- Authors claim the proposed method also works for continuous setting. This is not convincing, we all know the continuous setting can be more complex in many ways, do you have theoretical or empirical evidence to show it also works in continuous setting? If not should change the argument.
- first mentioned Figure 2 (which has MeDQN(U)) before section 3, and MeDQN(U) is not defined or explained until much later in the paper (4.3). Would be great if you add some additional explanation early on to help readers understand.
- End of page 5, why does functional regularization allows for optimizing θ in bigger parameter space? Is this knowledge from another paper?

**Strengths And Weaknesses:**

**Strengths**
- originality: the idea can be considered novel
- quality: hyperparameters discussed, extensive experiments and ablations, good explanation of things, many technical details reported, overall good quality.
- clarity: overall clear writing
- significance: the problem is important, reducing memory usage will allow more practical usage of DRL especially deploying to portable devices. The results seem to be good too.

**Weaknesses**
- Overall the paper is nicely written. My main concern is on related work, there is no comparison to any other techniques in the literature to address the forgetting problem. While the proposed method works very well compared to the naive baseline, how does it compare to other related works?
- Additionally, I have some questions:
- in the paper it is shown the proposed method has faster computation, authors argue this is due to less frequency in updating the networks. I'm not sure I am convinced because the proposed method takes multiple updates and computes additional loss which will bring in extra computation. More detailed explanation on why it is faster is required here.
- Some other questions and suggested change see next section.

**Summary**
Overall I like the paper, but I think there should be comparison to not just the very naive baselines, otherwise it is difficult to know how it compares to other related works. The other issues are fairly minor.

---

> ### Author Response · Authors · 2023-03-15
> **We thank you for the detailed feedback, suggestions, and comments.**
>
> We thank you for the detailed feedback, suggestions, and comments. In the following, we address your concerns one by one.
> 1. **Add more baselines that might help tackle the forgetting issue.** As we stated in the related work, most works in continual RL focus on incremental task learning, where tasks arrive sequentially with clear task boundaries. These methods require either clear task boundaries or a large buffer to learn each task one by one. Since we study memory-efficient (i.e., no large buffers) RL algorithms in single-task learning (i.e., no task boundaries), these methods are not applicable. In single RL tasks, the forgetting issue is underexplored and unaddressed, as the issue is masked by using a large replay buffer. As far as we know, no other RL algorithms in the literature have been shown to perform well under extreme memory constraints. Please point us to relevant literature if we have missed. Furthermore, in this work, we mainly focus on showing that 1) catastrophic forgetting exists in single RL tasks under memory constraints and 2) our methods can effectively reduce forgetting and maintain high performance under memory constraints, which we believe is a substantial contribution.
> 2. **Why MeDQN is faster to train with extra computation?** Our algorithms are computationally efficient due to less frequent updates of the neural networks. Specifically, the update frequency is controlled by $C_{current}$ (See Algorithm 1 Line 10). When $C_{current}$ is large enough (i.e., less frequent updates), our algorithms would still be faster despite additional computation due to taking multiple updates and computing the knowledge consolidation loss. Please see appendix C for hyper-parameter details.
> 3. **Other minor issues.** We have also updated Section 2 to explain MeDQN(U). The claim that functional regularization allows optimizing $\theta$ in bigger parameter space has been removed. The argument that our method works in the continuous setting has been updated to be more accurate.

---

> > ### Comment · Reviewer_xdVG · 2023-03-16
> > **Thanks for your reply**
> >
> > I thank the authors for the rebuttal. These are some valid points. Most of my concerns are addressed.
> >
> > On **Why MeDQN is faster to train with extra computation?**, is it correct that per-update-wise MeDQN(U) runs slower than baseline (due to additional loss), but when C_current is large, then the updates are less frequent, so MeDQN(U) essentially takes a smaller total number of updates compared to baseline, and as a result it runs faster?

---

> > > ### Author Response · Authors · 2023-03-16
> > > **Further clarification**
> > >
> > > Yes, exactly!
> > > Due to the forgetting issue, DQN needs more frequent updates to relearn from a large replay buffer. MeDQN needs less updates since we reduce the forgetting issue with knowledge consolidation.

---

> > > > ### Comment · Reviewer_xdVG · 2023-03-16
> > > > **Thanks!**
> > > >
> > > > Thanks for the clarification!
> > > >
> > > > Some further questions:
> > > > - Based on my current understanding, is it correct that your results show that, on MinAtar tasks that are tested, compared to the DQN baseline (with 1M buffer), with hyperpararmeters in Young & Tian (2019), MeDQN(R) only added the consolidation loss, and can achieve 1. similar performance; 2. with 10x smaller buffer; 3. with fewer updates; 4. with less computation time?
> > > >
> > > > - Table 1 and 2 captions didn't say which envs do we get the returns from. Are these from the 4 MinAtar games as well?

---

> > > > > ### Author Response · Authors · 2023-03-16
> > > > > **Answers to further questions**
> > > > >
> > > > > For the first question, yes, you are correct and thank you for summarizing the advantages of our methods here.
> > > > > For the second question, results in both tables are from Seaquest (MinAtar). We will updated the captions to clarify this. Thank you!

---

### Review · Reviewer_h7DU · 2023-03-09

**Summary Of Contributions:**

This paper studies training reinforcement learning agents in a low-memory budget setting. Specifically, The authors pointed out that the replay memory of off-policy RL algorithms is critical for preventing ‘catastrophic forgetting.’ In this paper, the authors aim to alleviate the ‘catastrophic forgetting’ without using a large replay buffer. To address the problem, the authors proposed to distill knowledge from a target Q network into a current Q-network. The proposed approach is evaluated on low-dimensional control games and a subset of Atari games. The experimental results show that the proposed approach achieves competitive performance with a much smaller replay buffer.

**Audience:**

Yes

**Broader Impact Concerns:**

The reviewer has no concern on the ethical implications of this paper.


**Claims And Evidence:**

No

**Requested Changes:**

The following changes might help strengthen the paper:

1. Better motivate the setting of low-memory RL. Giving more examples and discussion may help the reader better understand the motivation. For instance, in what kind of real world problems a large replay buffer would be an issue.

2. Reporting experimental results on more Atari games or environments with higher dimensional observation. For instance, embodied AI environments such as robo thor, have high-dimensional and multimodal observation. Environment with high-dimensional observation could better justify the need of memory-efficient RL.

3. Reporting the memory usage of the proposed methods and baselines.

4. It is unclear to the reviewer how to apply the proposed uniform state sampling to states that contain discrete tokens. For instance, suppose the observation consists of description in text. How to define the min and max? How to sample a reasonable pseudo-state? Please clarify.


**Strengths And Weaknesses:**

Strength:

1. The paper is well organized and clearly written. It is generally easy to follow and the readers are well-informed. The two examples in Figures (1) and (2) clearly illustrate the ‘catastrophic forgetting’ in reinforcement learning.

2. Memory constraint is a less-explored problem in our deep RL community. The findings presented by the paper may be valuable for future research on memory-efficient RL.

3. Related works on reducing ‘catastrophic forgetting’ are adequately cited and discussed.


Weakness:

1. The reviewer found the paper is not well-motivated. It is unclear to the reviewer in what real-world situation do we need to train an RL agent in an edge device. Giving more examples and discussion may help the reader better understand the motivation.

2. The reviewer has some concern on the proposed uniform state sampling method. It seems that the uniform state sampling method only works in environments with very low-dimensional numberical observation / state. However, when the observation is low-dimensional, why should one worry about the memory issue of the replay buffer?

3. The reviewer found the experimental results very limited. Besides the low-dimensional simple control tasks, the authors only report results on four out of more than fiftyAtari tasks. It is unclear if the proposed approach could scale to more complex tasks. In addition, the main topic of the paper is ‘memory efficiency’. However, there is no memory usage comparison provided in the experimental section.

---

> ### Author Response · Authors · 2023-03-15
> **We thank you for the detailed feedback, suggestions, and comments.**
>
> We thank you for the detailed feedback, suggestions, and comments. In the following, we address your concerns one by one.
> 1. **Reporting the memory usage of the proposed methods and baselines.** We have already reported the sizes of replay buffers for different algorithms. More details about memory usages are included in appendix E.
> 2. **How to apply the proposed uniform state sampling to states that contain discrete tokens?** First, we want to point out that dealing with discrete tokens (e.g., words of text) is not just the problem of our methods. It is the problem of all deep learning methods since neural networks cannot process words of text directly. If the number of discrete tokens is small, we can simply encode them with one-hot vectors. In natural language processing, words of text are usually transformed to [word embeddings](https://en.wikipedia.org/wiki/Word_embedding). This method can be combined with RL algorithms (including our methods) to process discrete tokens in states as well.
> 3. **Enhance the motivation: giving more examples and discussion about training RL agents with edge devices.** We have updated the introduction to add more examples of training RL agents in the real-world.
> 4. **It seems that MeDQN(U) only works in low-dimensional environments. However, why should one worry about the memory issue in this case?** In low-dimensional tasks, we do agree that the memory issue is less worrying in this case. In high-dimensional tasks, although MeDQN(U) does not perform as well as DQN or MeDQN(R), it still significantly outperforms DQN(S) (See Figure 5). That is, under extreme memory constraints (e.g., buffer size=32), MeDQN(U) would be a much better choice than DQN. We also expect that MeDQN(U) could inspire future research to come up with more memory-efficient algorithms.
> 5. **The experimental results are very limited.** Please check our general response.

---

> > ### Comment · Reviewer_h7DU · 2023-03-27
> > **Thanks for the response.**
> >
> > Thanks to the authors for the responses, which partially addressed my questions. The reviewer found the revised introduction better motivated the paper and the reported memory usage helpful. However, the reviewer still has concerns on the scalability to high-dimensional environments of the proposed method. Using only Atari games as a ‘high-dimensional’ environment is not convincing. Moreover, the authors mentioned that when the input is text, one could transform the text to word embeddings. However, it is still unclear to the reviewer how to define the min and max and how to sample a reasonable pseudo-state.

---

> > > ### Author Response · Authors · 2023-03-27
> > > **Further response**
> > >
> > > Thank you for taking the time to provide your feedback. We appreciate the opportunity to further clarify some of your concerns.
> > >
> > > Regarding the input size of high-dimensional tasks, we believe that Atari games with a input state size of 84*84, which is approximately 7K, are widely considered as high-dimensional tasks. As such, they are often used as benchmarks for many DQN-based algorithms [1-6]. However, if you have a different definition or criteria for high-dimensional tasks, we would be happy to hear your thoughts and learn more about it. Additionally, we would appreciate it if you could provide a specific task with a *discrete action space* that you consider to be qualified as a high-dimensional task, and we can compare DQN and our methods accordingly.
> > >
> > > Regarding text inputs, we understand your concerns and would like to provide a concrete example to clarify. Let's assume that we have a task called "fruit" and the agent receives text observations that contain the names of fruits only, such as "apple" and "banana". We can encode the texts into word embeddings using a language model $f$, such as BERT [7]. And by encoding texts into word embeddings, we can define element-wise max and min in the embedding space. For example, the embeddings for "apple" and "banana" could be [0, 0.5, -1.5] and [1.0, -1.0, 0.3], respectively. We initialize two vectors, $s_{LOW}$ and $s_{HIGH}$, with dimensions equivalent to the word embeddings, i.e. $s_{LOW} = [\infty, \infty, \infty]$ and $s_{HIGH} = [-\infty, -\infty, -\infty]$. When the agent receives "apple", we update $s_{LOW}$ and $s_{HIGH}$ as the following: $s_{LOW} = \min (s_{LOW}, s_1) = \min ([\infty, \infty, \infty], [0, 0.5, -1.5]) = [0, 0.5, -1.5]$ and $s_{HIGH} = \max (s_{HIGH}, s_1)  = \max ([-\infty, -\infty, -\infty], [0, 0.5, -1.5]) = [0, 0.5, -1.5]$. Next, when the agent receives ”banana”, we continue to update: $s_{LOW} = \min (s_{LOW}, s_2) = \min ([0, 0.5, -1.5], [1.0, -1.0, 0.3]) = [0, -1.0, -1.5]$ and $s_{HIGH} = \max (s_{HIGH}, s_2)  = \max ([0, 0.5, -1.5], [1.0, -1.0, 0.3]) = [1.0, 0.5, 0.3]$. Furthermore, we can sample pseudo-states uniformly from the interval $[s_{LOW}, s_{HIGH}]$. It's important to note that we do not sample text inputs, but rather pseudo-embeddings, skipping the encoding process. We hope this example addresses your concern.
> > >
> > > Thank you again for your feedback, and we are always striving to improve our methods and addressing your concerns or questions.
> > >
> > > [1] Mnih, V., Kavukcuoglu, K., Silver, D., Antonoglou, A. G. I., Wierstra, D., & Riedmiller, M. Playing Atari with Deep Reinforcement Learning.
> > > [2] Van Hasselt, H., Guez, A., & Silver, D. (2016, March). Deep reinforcement learning with double q-learning. In Proceedings of the AAAI conference on artificial intelligence (Vol. 30, No. 1).
> > > [3] Bellemare, M. G., Dabney, W., & Munos, R. (2017, July). A distributional perspective on reinforcement learning. In International conference on machine learning (pp. 449-458). PMLR.
> > > [4] Mnih, V., Kavukcuoglu, K., Silver, D., Rusu, A. A., Veness, J., Bellemare, M. G., ... & Hassabis, D. (2015). Human-level control through deep reinforcement learning. nature, 518(7540), 529-533.
> > > [5] Hessel, M., Modayil, J., Van Hasselt, H., Schaul, T., Ostrovski, G., Dabney, W., ... & Silver, D. (2018, April). Rainbow: Combining improvements in deep reinforcement learning. In Proceedings of the AAAI conference on artificial intelligence (Vol. 32, No. 1).
> > > [6] Li, Z., Li, Y., Zhang, Y., Zhang, T., & Luo, Z. Q. HyperDQN: A Randomized Exploration Method for Deep Reinforcement Learning. In International Conference on Learning Representations.
> > > [7] Kenton, J. D. M. W. C., & Toutanova, L. K. (2019). BERT: Pre-training of Deep Bidirectional Transformers for Language Understanding. In Proceedings of NAACL-HLT (pp. 4171-4186).

---

> ### Author Response · Authors · 2023-03-25
> **Official Comment by Authors**
>
> Dear Reviewer h7DU,
>
> Thank you so much for taking the time to review our paper. We sincerely appreciate your detailed feedback and appraisal of our work, and have carefully addressed your comments in our rebuttal response. As the author-reviewer discussion period is half-way through, we would be grateful if you could acknowledge receipt of our responses and let us know if they address your concerns. We are eager to engage in any further discussions if needed.

---

### Author Response · Authors · 2023-03-15
**General response to common concerns**

​​We notice that there is a common concern about experiments in high-dimension tasks.
First, we clarify why and how these MinAtar (Young & Tian, 2019) games are selected. For our work, MinAtar tasks are good benchmarks since they have high-dimensional image inputs and require a large replay buffer. Second, MinAtar captures the general mechanics of Atari 2600 games, which are suitable to show the effectiveness of our methods in complex games.
Currently, the platform consists of five environments: Seaquest, Asterix, Breakout, Space Invaders, and Freeway. The environments Seaquest, Asterix, and Space Invaders increase in difficulty upon certain game events to capture the curriculum learning aspect of the associated Atari games. Except for Breakout, each environment is partially observable due to things like the timing of object movement and the current difficulty level, which are not encoded in the state. Seaquest and Freeway present a greater exploration challenge than the other games. In Seaquest, learning to surface for air requires significant exploration. In Freeway, the exploration challenge is due to the sparsity of reward and the large number of coordinated actions necessary to reach it. Compared to training DQN in Atari games, It is much faster to train DQN in MinAtar games. Using MinAtar as a testbed, we can do extensive experiments and ablation studies, drawing more confident conclusions. In practice, we found it took much longer time to train in Freeway than other games and selected the other four games for testing.

To further evaluate our algorithm, we conducted experiments in Atari games (Bellemare et al., 2013) which are much more complex than MinAtar. Specifically, we selected five representative games recommended by Aitchison et al. (2022), including Battlezone, Double Dunk, Name This Game, Phoenix, and Qbert. All results are presented in appendix D & E. Overall, these results confirm that our algorithm is more memory efficient and computation efficient while achieving comparable or better performance than DQN. **Specifically, the memory size of a replay buffer is reduced from 7GB to 0.7GB without hurting the agent's performance.** We will release the source code for this experiment and all hyper-parameters as well.

---

### Decision · Action_Editors · 2023-04-03

**Recommendation:** Accept with minor revision

**Comment:**

Neural networks are state-of-the-art Q-function approximators in off-policy RL. When training these networks, experience replay is used to reduce forgetting and stabilize learning. This work proposes a new approach to experience replay, where the optimized Q network is regularized by the target Q network over samples from the experience replay buffer. The proposed approach is evaluated on several low-dimensional control problems and Atari games. The authors demonstrate that their approach requires a much smaller experience replay buffer to perform comparably to the original approach.

This work addresses an important problem and the reviewers recognize it. The approach is also simple and thus can be easily used by others. The reviewers disagree whether the paper demonstrated sufficiently good performance on more complex tasks. Specifically, two reviewers ask for:

* More Atari games, some of which are arguably hard for DQN-based algorithms.

* Continuous control tasks.

To reach a middle ground, I suggest the following:

* This paper will be **accepted with minor revision**.

* The authors will reduce the scope of their claims, including potentially changing the title, so that they better reflect the demonstrated value of the paper. I will review this and ask the reviewers to confirm. Please highlight the changes.

**Audience:**

Yes. This paper is for the RL community. Since off-policy RL is growing in importance, this work is timely.

**Claims And Evidence:**

Yes. The authors evaluate their approach empirically on 4 low-dimensional control problems and 9 Atari games.